# Bee Venom Melittin Protects against Cisplatin-Induced Acute Kidney Injury in Mice via the Regulation of M2 Macrophage Activation

**DOI:** 10.3390/toxins12090574

**Published:** 2020-09-06

**Authors:** Hyunseong Kim, Jin Young Hong, Wan-Jin Jeon, Seung Ho Baek, In-Hyuk Ha

**Affiliations:** 1Jaseng Spine and Joint Research Institute, Jaseng Medical Foundation, Seoul 135-896, Korea; biology4005@gmail.com (H.K.); vrt3757@gmail.com (J.Y.H.); poghkl@gmail.com (W.-J.J.); 2College of Korean Medicine, Dongguk University, 32 Dongguk-ro, Ilsandong-gu, Goyang-si 10326, Gyeonggi-do, Korea; Baekone99@gmail.com

**Keywords:** cisplatin, acute kidney injury, melittin, M2 macrophage

## Abstract

Inflammation is an essential biological response that eliminates pathogenic bacteria and repairs tissue after injury. Acute kidney injury (AKI) is associated with systemic and intrarenal inflammation as the inflammatory process decreases renal function and promotes progression to advanced chronic kidney disease. Macrophages are key mediators of the inflammatory response; their activation influences the immune system and may have various effects. Classically activated type I macrophages (M1) produce a variety of pro-inflammatory cytokines at the lesion site. However, anti-inflammatory type II macrophages (M2) are alternatively activated upon exposure to anti-inflammatory cytokines and are associated with wound healing and tissue repair following AKI. Here, we used melittin from bee venom to enhance the polarization of M2 macrophages and promote renal recovery after AKI. Melittin was administered to mice intraperitoneally for 5 days at various concentrations (10, 50, and 100 µg/kg); serum creatinine and blood urea nitrogen (BUN) levels were analyzed 72 h after cisplatin administration to confirm renal dysfunction. Melittin inhibited the cisplatin-induced increase in creatinine and BUN, an indicator of renal dysfunction. The expression of M1 markers (CD16/32) decreased significantly, whereas that of M2 markers (CD206, Arg1nase I) increased after melittin administration. Consistently, tubular necrosis was substantially reduced in melittin-treated mice. Thus, melittin alleviates cisplatin-induced AKI by regulating M2 macrophage expression.

## 1. Introduction

Cisplatin is a platinum-based chemotherapeutic agent that is effective in treating various forms of tumors [1]. However, the use of high-dose cisplatin for prolonged periods may cause renal toxicity, with acute renal failure being the most significant adverse effect [2]. Although the mechanisms involved in acute renal failure have not been clearly identified, the inflammatory response induced by macrophage activation is recognized as an important cause [3,4]. Recent experimental evidence has revealed that changes in the inflammatory response are involved in the pathophysiological mechanism underlying cisplatin-induced renal toxicity [5,6], and the need for further relevant studies has been emphasized. Previous studies have shown that bee venom and its active ingredient phospholipase A2 (PLA2) could increase the number of regulatory T cells (Tregs) to inhibit acute injury to the kidney and liver. Of the numerous candidates that may influence Treg populations, bee venom demonstrated excellent renal protective effects in a cisplatin-induced acute kidney injury model by effectively increasing the number of Tregs [7,8,9]. Moreover, PLA2, one of the main components of bee venom, was shown to control the expression of interleukin (IL)-10 and Tregs by binding to CD206, protect against cisplatin-induced acute kidney injury (AKI), and prevent inflammatory responses [7,10,11].

Here, we administered melittin, an important component of bee venom and a known PLA2 inducer, to mice with cisplatin-induced AKI to investigate its effects on macrophage-mediated inflammatory responses and renal function recovery. Macrophages, the key regulators of inflammation, are differentiated into type 1 (M1) and type 2 (M2) according to their function and morphology [12,13]. M1 macrophages increase inflammatory cytokine expression, typically promoting the expression of intermediate products of IL-1β, IL-6, nitric oxide, and reactive oxygen species. Conversely, M2 macrophages increase the expression of the resistin-like molecule α1 (Fizz1), Arg1nase I (Arg1), chitinase 3-like 3 (Ym1), IL-10, and mannose receptor C-type 1 (Mrc1; CD206) genes, which are associated with parasite infiltration, tissue remodeling, and immunoregulatory functions [14,15]. Accordingly, recent studies have investigated the differentiation and activation of macrophages in the early or developmental stages of inflammatory diseases by identifying the detailed mechanisms underlying macrophage polarization and the resulting therapeutic effects. Based on these results, we hypothesized that melittin may increase M2 macrophage activity and protect against cisplatin-induced AKI.

## 2. Results

### 2.1. Protective Effects of Melittin on Cisplatin-Induced AKI in Mice

Kidney function was monitored by analyzing the serum creatinine and blood urea nitrogen (BUN) levels 72 h after cisplatin administration. Changes in the creatinine and BUN levels in mice injected with phosphate-buffered saline (PBS; blank group), cisplatin, or melittin at various concentration in the presence of cisplatin are shown in Figure 1A,B. Melittin inhibited creatinine excretion and BUN increases in a dose-dependent manner. Furthermore, the survival rate was calculated until day 10 after cisplatin injection. In the Cis group, the first mouse died 78 h after cisplatin injection, and all mice had died at 180 h. In the groups receiving melittin, the first mouse died at 102 h, and one third of mice survived after all cisplatin-administered mice had died (Figure 1C). In addition, we checked the change of body weight in next day after melittin injection without cisplatin administration. The Cis group received injection of PBS for 5 days. The body weight did not change significantly compared with the Cis group in a dose dependent manner (Appendix A). We also confirmed the weight changing in PBS or melittin injected mice at 3 days after cisplatin administration. The melittin preinjected mice weighed slightly higher than mice whose injection of cisplatin was confirmed (Appendix A). These results confirm that melittin increased survival in mice by inhibiting cisplatin-induced kidney damage.

### 2.2. Reduction of Cisplatin-Induced Renal Tissue Damage by Melittin Administration

In each group, the hematoxylin and eosin (H&E)-stained kidney sections showed necrotic tubules indicating tubular injury (Figure 2A–E). The degree of the renal failure was determined by the morphological change, including brush border loss, red blood cell extravasation, tubule dilatation, tubule degeneration, tubule necrosis, and tubular cast formation. The blank group showed the normal pattern of renal tubules and glomeruli, whereas in the Cis group, cisplatin induced acute tubular damage and necrosis and significantly increased the renal damage score compared with that in the blank group. However, mice in the melittin-injected groups showed significantly lesser histological damage than the cisplatin group. Furthermore, melittin reduced cisplatin-induced renal tissue damage in a dose-dependent manner (Figure 2F). Therefore, melittin ameliorated cisplatin-induced renal tubular damage in mice.

### 2.3. Effects of Melittin on Pro- and Anti-Inflammatory Cytokine Production in Mice with AKI

To examine the effects of melittin on pro- and anti-inflammatory cytokine production after cisplatin administration in mice, we analyzed the expression of pro- and anti-inflammatory cytokines such as IL-6, IL-1β, and IL-10. The administration of melittin significantly prevented the cisplatin-induced expression of pro-inflammatory cytokines, such as IL-6 and IL-1β (Figure 3A,B), and increased expression of the anti-inflammatory cytokine IL-10 (Figure 3C) in mice with AKI.

### 2.4. Effects of Melittin on the Expression of M1 and M2 Macrophage-Associated Genes in Mice with AKI

We investigated the expression of the M1 and M2 macrophage-associated genes in each group via real-time polymerase chain reaction (PCR). In mouse kidneys, the relative mRNA expression of cyclooxygenase 2 (*COX-2*), an inflammatory gene, decreased significantly and in a dose-dependent manner in the melittin-treated groups compared to that in the cisplatin group (Figure 4A). In contrast, the expression of the anti-inflammatory cytokine IL-10 was significantly upregulated in mice treated with 50 and 100 µg/kg melittin compared to that in mice in the cisplatin group (Figure 4B). The macrophage receptor with collagenous structure (MARCO) gene is commonly used as a marker for M1 macrophages [16,17]. Here, MARCO expression decreased significantly in the 50 and 100 µg/kg melittin-treated groups compared to that in the cisplatin group (Figure 4C). Furthermore, Arg1 levels were significantly upregulated in a concentration-dependent manner in the melittin-treated groups (Figure 4D). These findings reveal that 50 and 100 µg/kg melittin were the most effective concentrations in reducing the inflammatory response, inhibiting M1 macrophage activation, and enhancing M2 macrophage activation in renal tissue.

### 2.5. Effects of Melittin on M1 and M2 Macrophage Populations in Mouse Splenocytes

To identify the phenotype of macrophage in splenocytes, we used flow cytometry to determine the relative percentage of CD16/32^+^ and Arg1^+^ cells, representing M1 and M2 macrophages, respectively, in each group (Figure 5A). We first observed that the population of F4/80^+^ macrophages increased dramatically in melittin-treated groups (Appendix A). Within this population, macrophage subsets were analyzed using CD16/32 as a marker of the M1 phenotype and Arg1 as a marker of the M2 phenotype. When treated with melittin at varying concentrations, the percentage of CD16/32^+^ cells in splenocytes decreased significantly and in a dose-dependent manner (Figure 5B). Conversely, cells treated with melittin at concentrations of 100 and 200 ng/mL showed higher populations of Arg1-positive cells (Figure 5C). We further confirmed the CD16/32 or Arg1 expression in splenocytes by immunocytochemistry (Appendix A). The F4/80^+^ macrophage was positively double stained with Arg1, but not stained with CD16/32, whereas, the Arg1 positive cells were not detected in the Cis group. In addition, we quantified the percentage of cells positive for CD16/32 and Arg1 in splenocytes treated with the major constituents of bee venom (melittin, PLA2, apamin) relative to that in the blank group using flow cytometry. The melittin significantly increased the Arg1+ cells among the components of bee venom (Appendix A). These results indicate that melittin induced macrophage proliferation and specifically promoted M2 rather than M1 differentiation.

### 2.6. Effect of Melittin on M2 Macrophage Infiltration into the Kidney

We carried out immunohistochemical staining to investigate the macrophage subsets in the renal tissue of mice treated with either melittin or PBS, in the presence or absence of cisplatin. The CD68-positive cells were not detected in the PBS group (Figure 6A), whereas we found that melittin induced macrophage infiltration in renal tissue and activated the CD206-positive M2 macrophages (also co-stained with CD68, Figure 6B). There were fewer CD16/32-positive M1 macrophages (also co-stained with CD68) in the melittin-treated groups exposed to cisplatin than in the Cis group (Figure 6C,D). Conversely, there were more CD206-positive M2 macrophages (also co-stained with CD68) in the melittin-treated groups with cisplatin administration (Figure 6C,D). Quantification of the fluorescent images also showed that melittin significantly inhibited CD16/32 expression in M1 macrophages and enhanced CD206 expression in M2 macrophages in the presence of cisplatin (Figure 6E,F). We further confirmed the expression difference in the protein level between groups of the product IL-1β and VEGF cytokines that occur in the activated state of M1 and M2 macrophages using immunohistochemistry. These stained images are shown in Appendix A. Therefore, these results indicate that melittin induced M2 macrophage infiltration into the kidney.

## 3. Discussion

Although cisplatin is a widely used first-generation chemotherapeutic agent with excellent anticancer efficacy, it may cause serious adverse effects, such as kidney injury, owing to its intrinsic toxicity [18,19]. Results from studies screening approximately 200 types of herbal extracts for the treatment of acute nephritis showed that bee venom increased immune cell numbers and inhibited nephritis [7,9]. Although bee venom is a toxin, numerous recent studies have demonstrated its beneficial effects, including anti-inflammatory, anticancer, anti-mutagenic, and anti-nociceptive activities [20,21,22,23,24]. Melittin, the main component of bee venom, has a linear structure comprising 26 amino acids and accounts for 50% of bee venom [25]. Numerous studies have shown that melittin causes pain and inflammation at high concentrations; however, at moderate concentrations, melittin actually exerts anti-inflammatory effects [26]. Moreover, melittin also plays anti-inflammatory roles in various disease models [27,28,29].

Acute nephritis is an inflammatory disease that occurs acutely in the kidney and presents risks of severe complications associated with the rapid deterioration of renal function [30]. Previous studies have shown that an increase in Tregs was induced by PLA2, one of the major components of bee venom, but not by melittin. PLA2 binds to CD206 on dendritic cells and upregulates Tregs and dendritic cells to increase IL-10 secretion. Melittin, which is the main component of bee venom, was shown to act as a PLA2 inducer in previous studies [31]. Therefore, we hypothesized that melittin may also be used to regulate CD206 expression and M2 macrophages. Our results showed that macrophage numbers increased dramatically in melittin-treated splenocytes. Notably, the proportion of M2 macrophages was specifically increased within total macrophage numbers. Therefore, melittin promoted an increase in M2 macrophages in renal tissue and infiltrated in the acute phases of renal inflammation.

The administration of melittin to mice with cisplatin-induced AKI induced M2 macrophage activation, and renal function recovery was achieved through decreases in creatinine and BUN levels, which are typical markers associated with renal function. Moreover, a survivability experiment showed that survival was clearly extended by the administration of melittin. Although the significant therapeutic effects of melittin have been demonstrated, it is important to determine the optimal non-toxic concentration to minimize adverse effects and maximize melittin’s anti-inflammatory properties. Recent studies have shown that the native form of melittin may cause lysis and toxicity in non-specific cells [32], and other studies have consistently attempted to reduce the toxicity of melittin through the mutation of fusion proteins and have investigated methods for the effective delivery of melittin to specific lesion sites [32,33]. Future studies should focus on the fundamental mechanism by which melittin induces the activation of M2 macrophages and the development of melittin as a safe therapeutic agent with increased anti-inflammatory actions and reduced toxicity.

## 4. Materials and Methods

### 4.1. Animals and Drug Treatments

All procedures were approved by the Jaseng Animal Care and Use Committee (approval no. 16-045) on september 27, 2019 Male C57 mice (7 weeks old, Daehan Bio Link, Chungbuk, Korea) were housed individually in standard cages under a 12-h light/dark cycle in a temperature-controlled environment (25 °C) with a humidity of 50% and had free access to food and water. Melittin (Sigma-Aldrich, St Louis, MO, USA) was administered daily via intraperitoneal injection at doses of 10, 50, or 100 µg/kg for 5 days. Blank and cisplatin groups were administered the same volume of PBS in the same manner. The day following the final melittin injection, all mice other than those in the blank group were administered cisplatin (15 mg/kg). The mice were sacrificed 72 h after cisplatin administration, and the serum was collected. The kidneys were harvested and incubated either in liquid nitrogen before performing ELISA or in 4% paraformaldehyde (PFA, Biosesang, Seongnam, Korea) for H&E staining. To investigate mouse survival, a survival-prolonging effect was checked every 6 h for 10 days in the groups receiving 100 µg/kg melittin as compared with that in the Cis group receiving the cisplatin, and deaths were recorded. Samples were then divided into 5 groups (*n* = 6–8/group): Blank group (blank); no-treatment, Cisplatin group (Cis); 15 mg/kg cisplatin only injection, Melittin 10 (M10 + Cis); 10 µg/kg melittin + cisplatin injection, Melittin 50 (M50 + Cis); 50 µg/kg melittin + cisplatin injection, Melittin 100 (M100 + Cis); 100 µg/kg melittin + cisplatin injection. We have outlined our experimental procedures in more detail in a timetable, which we have added in Scheme 1.

### 4.2. Renal Function Analysis

Renal function was assessed by measuring the serum creatinine and BUN levels. Briefly, blood samples were obtained via intracardiac puncture and centrifuged at 1000× *g* for 10 min at room temperature to collect serum. The creatinine and BUN levels were measured using an automated dry chemistry analyzer for veterinary analysis (FUJI DRI-CHEM 7000i, Tokyo, Japan).

### 4.3. Morphological Assessment

Six mice from each group were sacrificed via cardiac perfusion for H&E staining and immunohistochemistry. Briefly, the mice were perfused with 0.9% normal saline and 4% PFA in PBS (pH 7.4). The kidneys were dissected and post-fixed overnight in 4% PFA at 4 °C. Samples were dehydrated in a graded series of ethanol, embedded in paraffin blocks, and cut across the coronal plane to obtain 10-µm sections. H&E staining was performed on the center site to examine kidney tubular damage 6 h after cisplatin injection according to the standard protocol [7]. The stained sections were imaged using an inverted microscope (Nikon, Tokyo, Japan). Three observers who were blinded to the treatment scored the degree of morphological involvement in renal failure: brush border loss, red blood cell extravasation, tubule dilatation, tubule degeneration, tubule necrosis, and tubular cast formation. Renal tubular injury was evaluated on a scale of 0 to 4, according to the percentage of cortical tubules showing epithelial necrosis: none (0); mild (<10%; 1 point), moderate (10–25%; 2 point), severe (25–75%; 3 point), to very severe (>75%; 4 point) [32]. Each parameter was examined on at least five different animals.

### 4.4. ELISA

Cytokines involved in the inflammatory response (IL-6, IL-1β, and IL-10) were analyzed using ELISA. The samples were homogenized in radioimmunoprecipitation assay (RIPA) buffer (GenDEPOT, Barker, TX, USA) containing a proteinase inhibitor (Millipore) using a taco^TM^ Prep Bead Beater (GeneReach, Taichung, Taiwan) and centrifuged at 13,000 rpm at 4 °C for 10 min. Protein concentration was quantified using a bicinchoninic acid (BCA) protein assay kit (Thermo Fisher Scientific, Waltham, MA, USA). The cytokines were examined using ELISA kits (BD Biosciences, Franklin Lakes, NJ, USA and Abcam, Cambridge, UK) according to the manufacturer’s protocol.

### 4.5. Real-Time PCR

We quantified changes in the expression of M1 and M2 macrophage-associated genes after melittin administration via real-time PCR. Briefly, one entire kidney in right side was homogenized in RLT lysis buffer using the taco™ Prep Bead Beater system (Taco, Taichung, Taiwan). After complete homogenization, we centrifuged kidney lysates for 3 min at 17,000 rpm (table top centrifuge at room temperature). We then transferred the supernatant as much as possible into column supplied in commercial AllPrep DNA/RNA/Protein Mini Kit (Qiagen, Hilden, Germany). Total RNA and protein were isolated according to the manufacturer’s instructions. cDNA was synthesized using random hexamer primers and Accupower RT premix (Bioneer, Daejeon, Korea). All primer pairs were designed using the University of California Santa Cruz (UCSC) Genome Bioinformatics and the National Center for Biotechnology Information (NCBI) databases (Primer-Blast, National Institutes of Health; Table 1). Real-time PCR was performed using a SYBR green supermix (Bio-Rad, Hercules, CA, USA) on a CFX connect Real-Time PCR Detection System (Bio-Rad). Each real-time PCR was performed in triplicate at least. The expression of each target gene was normalized to that of β-actin and is reported as the fold-change relative to that in the cisplatin group.

### 4.6. Immunohistochemistry

Immunohistochemistry was performed to analyze macrophage phenotypes in the kidney after the administration of melittin and cisplatin. Sections were permeabilized with 0.2% Triton X-100 in PBS for 5 min, washed with PBS, and blocked with 10% normal goat serum in PBS for 1 h. Next, fluorescent dye-conjugated antibodies, mouse anti-monocyte/macrophage CD68 (1:500, fluorescein isothiocyanate, FITC, Biolegend, San Diego, CA, USA), M1 macrophage CD16/32 (1:500, phycoerythrin, PE, BD Biosciences), and M2 macrophage CD206 (1:500, allophycocyanin, APC, BD Biosciences) were incubated for 2 h at room temperature. After incubation, the sections were washed thrice for 5 min with PBS. The cell nuclei were stained with 4′,6-diamidino-2-phenylindole (DAPI, 1:1000, TCI, Tokyo, Japan). Macrophage subsets was quantified by manually counting the total number of cells that appear brighter signal in response to CD16/32 or CD206 antibodies by the number of CD68^+^ cells in 10 consecutive high-power fields (HPF); 10HPF means the sum of 10 slides in which the cells expressing CD68 are seeing their highest average in each group (*n* = 6/group). The stained tissue sections were imaged 100 × magnification on a confocal microscope (Eclipse C2 Plus, Nikon, Tokyo, Japan).

### 4.7. Preparation of Murine Splenocytes

Mice were sacrificed via CO2 asphyxiation and placed on a clean dissection board. The abdominal cavity was immediately incised, and the spleen was removed. The isolated spleen was placed on a 40-μm cell strainer (Falcon, NY, USA) over a 6-well plate containing PBS. After centrifuging at 1500 rpm for 3 min, red blood cells (RBCs) were removed by incubating with RBC lysis buffer (Invitrogen, Carlsbad, CA, USA) for 15 min at room temperature. After washing, the splenocytes were incubated in RPMI (+10% FBS, 1% P/S, 2 μg/mL CD28) with melittin at 50, 100, or 200 ng/mL for 2 days. Flow cytometry was used to analyze the macrophage population from splenocytes treated with melittin. Briefly, the cells were incubated with the relevant antibodies (CD16/32, Arg1nase1 and F4/80) for 30 min at 4 °C and fixed with cell fixation buffer (BD Biosciences) for 10 min. Finally, 500 µL of PBS was added for fluorescence-activated cell sorting (FACS). We have outlined our experimental procedures in more detail in a timetable, which we have added in Scheme 2.

### 4.8. Statistics

All results are expressed as the means ± standard error of the mean (SEM). Single comparisons between two groups were made using Student’s *t*-tests, and multiple comparisons among three or more groups were analyzed via one-way analysis of variance (ANOVA) with Tukey’s post-hoc test. Significant differences between survival curves were analyzed using a log-rank (Mantel–Cox) test. Differences were considered statistically significant if *p* < 0.05.

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
