# Peer review of "Bee Venom Melittin Protects against Cisplatin-Induced Acute Kidney Injury in Mice via the Regulation of M2 Macrophage Activation"

_toxins, 2020, doi:10.3390/toxins12090574_

Round 1
Reviewer 1 Report
Paper prepared with high care, results presented clear, methodology correct detection techiques adequate.
Author Response
We were very pleased with the favorable reviews on our manuscript. We thank you for your thoughtful and helpful comments. We have extensively refined our presentation of the manuscript, figures, and references in more detail. Below, please find our point-by-point responses to all raised queries.
Kind regards,
Reviewer 2 Report
The authors explore the beneficial effects of melittin on cisplatin-induced kidney tubular injury, and link this to M2 polarization.
Some issues need to be addressed.
The dose of 15 mg/kg of cisplatin is lower than commonly used 20 or 25 mg/kg. Any reason for this.
Was the proportion of mice dying related to the melittin dose and can this be shown.
The histology based images are of low magnification and at least an inset of high magnification is required. In fig 2, tubular necrosis (which was used to determine the degree of tubular injury) cannot be properly identified at the current magnification. In fig 6, localization of macrophages is unclear, and moreover, the cells seem to be quite unevenly distributed, which is somewhat unusual.
Is the overall number of kidney macrophages similar between groups and can this be quantified and shown with F4/80 staining.
It is possible that the M2/M1 proportion is increased after melittin because of less tubular injury. Can the authors disprove this. If not, the conclusion has to be less definite regarding the mechanism of melittin. On a related note, please separate and clarify the data on splenocytes, as to cells that were obtained from mice treated with melittin, and isolated splenocytes that were then treated with melittin. Is it possible that Tregs are increased and these promote M2 polarization.
Author Response
Dear Reviewer
We were very pleased with the favorable reviews on our manuscript. We thank you for your thoughtful and helpful comments. We have extensively refined our presentation of the manuscript, figures, and references in accordance with your suggestions and the changes are marked in RED color in the revised manuscript. Below, please find our point-by-point responses to all raised queries.
Kind regards,
<Reviewer 2>
The authors explore the beneficial effects of melittin on cisplatin-induced kidney tubular injury, and link this to M2 polarization.
Some issues need to be addressed.
- The dose of 15 mg/kg of cisplatin is lower than commonly used 20 or 25 mg/kg. Any reason for this. Was the proportion of mice dying related to the melittin dose and can this be shown.
-We agree with the reviewer’s comment. We initially tested a broad range (15, 20 or 25 mg/kg) of cisplatin concentration for examine the optimal concentration to injection based on the references associated with induction of acute kidney injury (Perše et al., BioMed Research International, 2018. DOI: 10.1155/2018/1462802. Topcu-Tarladacalisir et al., Renal failure, 2016. DOI: 10.1080/0886022X.2016.1229996. Ma et al., Am J Transl Res, 2015.). However, when we injected 20 or 25 mg/kg of cisplatin into the mice, we found dying mice after night. Therefore, we choose the 15 mg/kg as final concentration of cisplatin for injection. In addition, we checked the change of body weight in next day after melittin injection without cisplatin administration (Supplementary Figure S1A). The control group received injection of PBS for 5 day. The melittin group received with different concentration. And, we also confirmed the weight changing in PBS or melittin injected mice at 3 days after cisplatin administration (Supplementary Figure S1B). The weight gain ratio was calculated as below.
Figure S4A. (D4/D0 × 100)-100 (%), Figure S4B. (D8/D5× 100)-100 (%)
- The histology based images are of low magnification and at least an inset of high magnification is required. In fig 2, tubular necrosis (which was used to determine the degree of tubular injury) cannot be properly identified at the current magnification. In fig 6, localization of macrophages is unclear, and moreover, the cells seem to be quite unevenly distributed, which is somewhat unusual.
- We appreciate the reviewer’s comment, and have now added high magnification image corresponding to its low image of each group in the Figure 2. And, we have corrected the result description in detail of Figure 2 on (Lines 87) as follows:
“In each group, the hematoxylin and eosin (H&E)-stained kidney sections showed necrotic tubules indicating tubular injury (Figure 2A-E). The degree of the renal failure was determined by the morphological change, including brush border loss, red blood cell extravasation, tubule dilatation, tubule degeneration, tubule necrosis, and tubular cast formation. The blank group showed the normal pattern of renal tubules and glomeruli, whereas in control group, cisplatin induced acute tubular damage and necrosis and significantly increased the renal damage score compared with that in the blank group. However, mice in the melittin-injected groups showed significantly lesser histological damage than the cisplatin group. Furthermore, melittin reduced cisplatin-induced renal tissue damage in a dose-dependent manner (Figure2F). Therefore, melittin ameliorated cisplatin-induced renal tubular damage in mice.”
- Actually, we performed double staining with CD68 (a specific marker of macrophage and monocyte) and CD16/32 (M1 macrophage) or CD206 (M2 macrophage) to confirm even more clearly macrophage types increased from CD68-positive total macrophage in the renal tissue of each group. In the revised manuscript, we changed the images in Figure 6 and the result description in the result section (Line 174) as follow;
“We carried out immunohistochemical staining to investigate the macrophage subsets in the renal tissue of mice treated with either melittin or PBS, in the presence or absence of cisplatin. The CD68-positive cells were not detected in PBS group (Figure 6A). Whereas, we found that melittin induces macrophage infiltration in renal tissue and activates the CD206-positive M2 macrophages (also co-stained with CD68, Figure 6B). There were fewer CD16/32-positive M1 macrophages (also co-stained with CD68) in the melittin-treated groups exposed to cisplatin than in the control group (Figure 6C, D). Conversely, there were more CD206-positive M2 macrophages (also co-stained with CD68) in the melittin-treated groups with cisplatin administration (Figure 6C, D). Quantification of the fluorescent images also showed that melittin significantly inhibited CD16/32 expression in M1 macrophages and enhanced CD206 expression in M2 macrophages in the presence of cisplatin (Figure 6E, F). Therefore, these results indicate that melittin induced M2 macrophage infiltration into the kidney.”
- And, we examined the distribution of M1 and M2 macrophages in immunostained images.
macrophages are rarely found and usually clustered in damaged kidney. Previous studies in kidney disease have shown that monocytes/macrophages were almost exclusively localized in areas of severe tissue damage and infiltrated renal tissue as a result of tissue damage (Lan et al., Kidney international, 1995. DOI: 10.1038/ki.1995.347. Baek et al., Frontiers in Physiology, 2019. DOI: 10.3389/fphys.2019.01016).
- Is the overall number of kidney macrophages similar between groups and can this be quantified and shown with F4/80 staining.
- We quantified the total number of cells positive for CD16/32 and CD206, double stained to CD68-postivie cells in 10 consecutive high-power fields (HPF). 10HPF means the sum of 10 slides in which the cells expressing CD68 are seeing their highest average in each group. We have now added this quantification content in the method section of revised manuscript (Line 316).
“Macrophage subsets was quantified by manually counting the total number of cells that appear brighter signal in response to CD16/32 or CD206 antibodies by the number of CD68 + cells in 10 consecutive high-power fields (HPF). 10HPF means the sum of 10 slides in which the cells expressing CD68 are seeing their highest average in each group (n=6/group). The stained tissue sections were imaged 100 ×magnification on a confocal microscope (Eclipse C2 Plus, Nikon, Tokyo, Japan).”
- It is possible that the M2/M1 proportion is increased after melittin because of less tubular injury. Can the authors disprove this. If not, the conclusion has to be less definite regarding the mechanism of melittin. On a related note, please separate and clarify the data on splenocytes, as to cells that were obtained from mice treated with melittin, and isolated splenocytes that were then treated with melittin. Is it possible that Tregs are increased and these promote M2 polarization.
- We apologize for not providing sufficient clarification. We have now separate and clarify the data on splenocytes
1) It is possible that the M2/M1 proportion is increased after melittin because of less tubular injury. Can the authors disprove this. If not, the conclusion has to be less definite regarding the mechanism of melittin.
- In previous study, we have confirmed that Treg infiltrate into the kidney by cisplatin administration. Treg migrate to the kidney in early time (after 6 hours) and block to M1 macrophage infiltration. We conducted an experiment in a short time after cisplatin administration in order to investigate the initial immune response occurring in the kidney after cisplatin administration. reference
(Lee et al., Kidney International, 2010. DOI: 10.1038/ki.2010.139)
2) please separate and clarify the data on splenocytes
- Thank you for highlighting these misses. In the revised manuscript, we have corrected drug treatment in splenocytes in detail and added the experimental scheme in the methods section (Line. 324) as follows:
“Mice were sacrificed via CO2 asphyxiation and placed on a clean dissection board. The abdominal cavity was immediately incised, and the spleen was removed. The isolated spleen was placed on a 40-μm cell strainer (Falcon, NY, USA) over a 6-well plate containing PBS. After centrifuging at 1,500 rpm for 3 min, red blood cells (RBCs) were removed by incubating with RBC lysis buffer (Invitrogen, Carlsbad, CA, USA) for 15 min at room temperature. After washing, the splenocytes were incubated in RPMI(+ 10% FBS, 1% P/S, 2 μg/ml CD28) with melittin at 50, 100 or 200 ng/ml for 2 days. Flow cytometry was used to analyze the macrophage population from splenocytes treated with melittin. Briefly, the cells were incubated with the relevant antibodies (CD16/32, Arginase1 and F4/80) for 30 min at 4°C. And, fixed with cell fixation buffer (BD Biosciences) for 10 min. Finally, 500 µl of PBS was added for fluorescence-activated cell sorting (FACS).”
Scheme 2. Schematic illustration of the experimental timeline. Mouse primary splenocytes was isolated from spleen of C57BL/6 mouse. Cells were seeded at a density of 4× 105 cells/well in 24 well plate. After 2 hours, splenocytes were treated with different concentration (50, 100, 200 ng/ml) of melittin for 2 days. And, we evaluated the effect of melittin on macrophage polarization in mouse splenocytes using FACS analysis and immunocytochemistry.
3) Is it possible that Tregs are increased and these promote M2 polarization.
- When we first designed this experiment, we hypothesized that melittin have protective effect in AKI model through M2 macrophage activation, inspired by the ideas from reference 7 and 10. Previous studies have shown that the bee venom and PLA2 inhibit cisplatin-induced acute kidney injury by modulating regulatory T cells by the CD206 mannose receptor. And, melittin is already known as head inducer of PLA2. we confirmed anti-inflammatory cytokines (IL-10) release from the activation of M2 macrophage in AKI model injected with melittin, although the precise mechanisms regulating this event remain unclear. These contents are summarized below.

Reviewer 3 Report
This is an interesting study, but has major drawbacks.
Critique:
- Most significant problem is that this study does not have Melittin only controls.
- Introduction: Please check reference numbers. Ref#9 is not the correct reference for " Of the numerous candidates that may influence Treg populations, bee venom demonstrated excellent renal protective effects in a cisplatin-induced acute kidney injury[9] model by effectively increasing the number of Tregs [10].
- The authors keep talking about PLA2 being activated by bee venom and assume that Melittin activates PLA2 also. No references are provided. The authors have not shown that Melittin increases IL10 by activating PLA2, no knockout studies of PLA2 have been shown.
- The results are lacking details.
- All graphs and figures need to be relabeled and “melittin only” group data needs to be added.
- Treatment paradigm has to be included in the results.
- Figure legends need to be expanded.
- In Fig. 1C the authors do not say what dose of melittin is used for survival studies and how many survive. What happens to the group treated with Melittin alone? Is there a dose response?
- 2, there is no explanation on how the scoring is performed by individual observers of renal damage. How is the data quantified?
- Figure 3 which tissue are they using to estimate the amount of cytokine? How is the tissue processed?
- Fig 4. How are the authors coming up with MARCO? Where are the references? Where is the Immunohistochemistry data to support the RNA data? This data indicates that there is no significant increase in MARCO in cisplatin treated group from control/blank. Gene expression for IL-10 is well below 1 for cisplatin, while Melittin seems to significantly increase IL10 expression while remaining well below the basal level. This data is hard to believe unless followed by increases in protein expression as shown by immunohistochemistry.
- 5 does not make sense at all. Are these groups treated with cisplatin? If so, where is the cisplatin data? If not what is the treatment paradigm for Melittin treatment and what is the rationale for this experiment. The Figure legend clearly indicates that “ ***p<0.001 vs the cisplatin group were analyzed via one-way ANOVA and Tukey’s post-hoc test.”
- Figure 6: what is being quantitated? The spots of fluorescence look as if the slides were not washed well. Scale bar in the figure says 50µm while the legend indicates 20µ How many 10HPF’s were evaluated? What was the N for each experiment? The images are not sharp and difficult to see.
- This manuscript does not provide data that is robust enough to validate the hypothesis of Melittin causing M2 macrophage activation in the kidneys.
- Discussion: the authors circle back to PLA2 without any evidence. They seem to be heavily reliant om reference #10. The experiments performed are similar, albiet with a different component of bee venom.
- Overall this could make an interesting manuscript if the authors made major revisions and performed additional experiments.
Author Response
Dear Reviewer
We appreciate your favorable review regarding our manuscript. We sincerely appreciate all valuable comments and suggestions, which helped us to improve the quality of the article. Our responses to the Reviewers’ comment are described below in a point-to-point manner. Appropriated changes, suggested by the Reviewers, has been introduced to the manuscript (highlighted within the manuscript). We hope that our manuscript will be acceptable for publication in Toxins.
Kind regards,

Round 2
Reviewer 3 Report
The authors have made significant changes in the manuscript.
Minor comments:
- In all Figures, please rename “Control” as cisplatin and “M10” as “M10+Cisplatin” etc. to clarify the groups.
Author Response
We thank you for your thoughtful and helpful comment.
The group names were changed according the comment and highlighted in RED in revised manuscript.
- blank
- Cis : cisplatin only
- M10+Cis : melittin 10 μg/kg + Cis
- M50+Cis : melittin 50 μg/kg + Cis
- M100+Cis : melittin 100 μg/kg + Cis
